# The self-reference memory bias is preceded by an other-reference bias in infancy

Charlotte Grosse Wiesmann [1,2,3] ✉, Katrin Rothmaler [1], Esra Hasan[1], Kathrine Habdank[2], Chen Yang [1], Emanuela Yeung[2] & Victoria Southgate[2]

One of the most established biases in human memory is that we remember information better when it refers to ourselves. We investigated the development of this *self-reference effect* and its relationship with the emergence of a self-concept. We presented 18-month-old infants with objects that were assigned either to them, or to another agent. Infants were then tested on their memory for the objects by presenting them with an image of each object, alongside a modified version of it. Mirror self-recognition served as an index of self-concept emergence. Infants who recognize themselves in the mirror remember objects assigned to themselves better than those assigned to the other. In contrast, non-self-recognizers only remember the objects assigned to the other rather than themselves. This difference is not explained by differences in infants' age or inhibitory abilities. This suggests that the self-reference effect emerges with the development of self-concept in the second year. Prior to the emergence of a self-concept, however, infants instead seem to exhibit an *other-reference effect*. This reversal of the classic self-reference effect suggests that early in life, when infants are heavily reliant on others for information, they may be biased towards encoding the world as it relates to others.

In human cognition, the concept of self plays a central role in guiding how we process our environment. As adults, we remember things related to ourselves better than those concerning others. For example, when asked to list famous people who share our first name, we typically recall more people than we would for a different name, even if these names are equally familiar[1,2]. This *self-reference effect* confers a robust memory advantage for self-related content[3], and it is proposed that because self-related information has high goal-relevance, items that have relevance for the self are prioritized in attention and subsequent encoding, resulting in better memory[4,5]. Beyond attention allocation, there are several other theoretical explanations for why self-relevant information is remembered better, including the possibility for more elaborative encoding and better organization of items within the framework of rich conceptual self-knowledge[6–8]. This extensive body of stored self-knowledge is thought to provide a way of scaffolding incoming information in a way that is not possible when that

information is related to others. If this is so, it raises the question of how this memory bias develops in childhood, assuming such a body of self-knowledge takes time to construct. Furthermore, the construction of knowledge about the self would seem to require a prior concept of self for the child to evaluate incoming information as self-related. Given that a concept of self develops slowly with clear evidence only towards the end of the second year of life, how is memory structured before the emergence of a concept of self?

While the ontogeny of the self-reference effect is unknown, several studies have shown that preschool children already show a memory benefit for content encoded in relation to the self[5,9,10]. For example, 4- to 6-year-old children were better at recalling objects that they had previously been asked to evaluate in relation to themselves (e.g., do you like apples?) compared to objects evaluated in relation to another child (e.g., does he/she like apples?)[5]. They even remembered objects better if asked whether they were shown to the left or right of

[1]Research Group Milestones of Early Cognitive Development, Max Planck Institute for Human Cognitive and Brain Sciences, Leipzig, Saxony, Germany. [2]Center for Early Childhood Cognition, Department of Psychology, University of Copenhagen, Copenhagen, Denmark. [3]Cognitive Neuroscience Lab, Department of Liberal Arts and Sciences, University of Technology Nuremberg, Nuremberg, Germany. ✉e-mail: wiesmann@cbs.mpg.de

their own photograph, as opposed to another child's picture. Building on ownership as a salient dimension of categorizing objects in relation to the self from early in development, a study showed that 3-year-old children remembered items sorted into their shopping basket better than those assigned to another child[9]. Taken together, these studies suggest that the self-reference effect is present from at least the preschool years.

In the second year of life, infants undergo important developments in their self-awareness that we reasoned may contribute to the development of the self-reference effect[11], either because of increased attention to items that may now be encoded as self-relevant, or because of differences in organization of incoming information in relation to the emerging self-concept[8]. Indeed, a large body of research suggests that during this period, infants develop a conceptual representation of themselves as individuals that are similar to, but independent from, other people[12,13]. The most established measure of the emergence of a self-concept is the mirror self-recognition test[14]. In this test, a colored mark is applied to the child's face and the child is then confronted with their own mirror image. From around 18 months, infants begin to touch the mark on their face, indicating that they recognize themselves in the mirror[14,15]. Mirror self-recognition correlates not only with toddlers' use of verbal self-reference like *I, me, mine*[16,17], but also with the emergence of self-conscious emotions such as embarrassment[12,17,18], the tendency to align the self with others[15,19,20], and greater functional connectivity between brain regions implicated in self-related processing[21]. This close link between mirror self-recognition and other abilities indicating an understanding of the self in relation to others[12,13,15], corroborates the emergence of a self-concept in this period and suggests the mirror self-recognition test is a valuable indicator of this development. Based on these findings, we hypothesized that the memory advantage for self-related information (i.e., the self-reference effect) may emerge in this period, in relation to infants' emerging self-concept.

To test this, we presented 18-month-olds with novel objects that were either introduced as for them, or for a puppet. We then tested their memory for these objects. In addition, infants participated in a mirror self-recognition test. We preregistered the hypothesis that infants who recognized themselves in the mirror would show a memory benefit for the self-assigned objects, whereas toddlers who did not recognize themselves would not. The relevance of the objects was established through an ownership manipulation (self versus other), building on findings that an understanding of ownership is present from early in development and a prominent means of highlighting a relation to the self[13,22,23].

Specifically, infants (N = 73) were presented with two boxes placed on a table in front of them (see Fig. 1). One of these boxes was introduced as their own. They were told that toys placed in this box would belong to them and that they would be able to play with them later. The other box was introduced as the puppet Sven's box. In the object encoding phase, the experimenter then presented one unknown object after another, alternately introducing them as for the child or for Sven, and then placed the object into the respective box and closed the lid. Next, infants' memory for the self-assigned and for the other-assigned objects was tested by presenting each object on a screen, alongside a second similar but slightly modified object (see Supplementary Information, SI, Figure S1). With an eye-tracker we measured the duration of infants' gaze to the previously seen, *familiar* object compared to the modified, *novel* object. Building on classic preferential looking paradigms[24,25], we reasoned that if, across object pairs, infants showed a systematic preference either for the familiar object or the novel object, this would indicate that they were able to differentiate them and thus remembered which objects they had previously seen. As memory in infants may manifest as either a familiarity or a novelty preference depending on age and task[24], we replicated the observed familiarity preference in a second sample of toddlers aged 20-40 months (N = 82, Experiment 2). Both experiments were preregistered (details see "Methods"), and the procedures and analyses follow these preregistrations unless explicitly stated otherwise.

In this work, we show a transition in infants' memory with the emergence of mirror self-recognition. Across both experiments, infants who recognized themselves in the mirror tended to remember the self-assigned objects better than the other-assigned objects. In contrast, prior to mirror self-recognition, infants showed better memory for the other-assigned objects. This indicates that, before the emergence of a self-concept, infants show a reversal of the classic self-reference effect, namely, an *other-reference effect*.

## Results

### Experiment 1: memory for self- versus other-owned objects by mirror self-recognition

To test our prediction, we calculated a differential looking score (DLS) as the normalized difference between looking duration to the familiar and the novel object. We then tested whether toddlers' memory for the self-owned and the other-owned objects depended on their mirror self-recognition with a mixed Bayesian ANOVA with default priors including the factors *mirror self-recognition* (recognizers vs. non-recognizers) and *ownership* (self-owned vs other-owned). As predicted, this yielded evidence for an interaction between *mirror self-recognition* and *ownership*

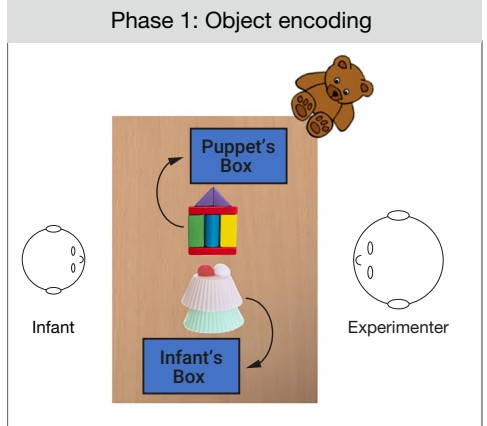
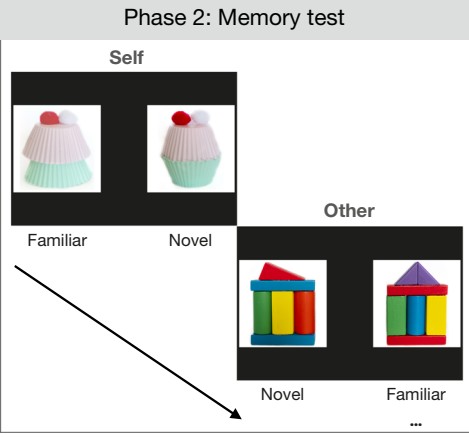

**Fig. 1 | Overview of the experimental set-up.** In the object encoding phase, infants were presented with a series of objects, alternately assigned to themselves, or to a puppet. In the subsequent memory test phase, these objects were presented on a screen alongside a second similar but modified object, and infants' memory for the objects was tested with a preferential looking paradigm (details see text). Teddy bear by Clker-Free-Vector-Images via Pixabay.

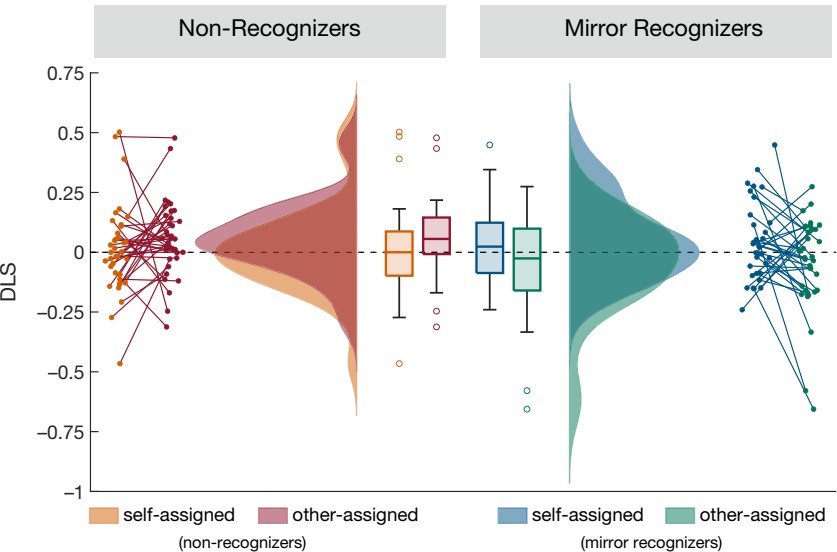

**Fig. 2 | Differential Looking Score (DLS) for children who did (right) or did not (left) pass the mirror self-recognition test.** Non-recognizers (N = 37) only remembered other-assigned but not self-assigned objects (as shown by a positive DLS, reflecting a preference for the familiar objects), whereas mirror-recognizers (N = 34) tended to remember the self-assigned objects better than the other-assigned objects. On each box, the central mark indicates the median, and the bottom and top edges of the box the 25th and 75th percentiles. The whiskers extend to the most extreme data points without outliers, and the outliers are plotted individually as circles.

(Bayes Factor $BF_{10}$ = 4.28, error = 1.71%, $\eta_p^2$ = 0.079, Fig. 2). For mirror recognizers (N = 34), there was anecdotal evidence for a higher DLS for self-owned than for other-owned objects ($\underline{DLS}_{self}$ = 0.044, $std_{self}$ = 0.163, $\underline{DLS}_{other}$ = −0.049, $std_{other}$ = 0.197, Wilcoxon signed rank test for higher DLS in self than in other: $BF_{10}$ = 2.685, W = 407, $\hat{R}$ = 1.001, $r_{rb}$ = 0.368). Mirror non-recognizers (N = 37), in contrast, showed a higher DLS for other-owned objects ($\underline{DLS}_{self}$ = 0.012, $std_{self}$ = 0.186, $\underline{DLS}_{other}$ = 0.057, $std_{other}$ = 0.155, strong evidence against a higher DLS in self than other according to Wilcoxon signed rank test: $BF_{10}$ = 0.077, W = 227, $\hat{R}$ = 1.001, $r_{rb}$ = −0.318). There was no main effect of *ownership* in the whole sample ($BF_{10}$ = 0.23, error = 1.503%, $\eta_p^2$ = 0.01), nor a main effect of *mirror self-recognition* ($BF_{10}$ = 0.371, error = 1.146%, $\eta_p^2$ = 0.021). When collapsing mirror recognizers and non-recognizers, there was evidence against a difference in the DLS between self-owned and other-owned objects ($\underline{DLS}_{self}$ = 0.027, $std_{self}$ = 0.172, $\underline{DLS}_{other}$ = 0.002, $std_{other}$ = 0.185, two-tailed Wilcoxon signed rank test: $BF_{10}$ = 0.159, W = 1388, $\hat{R}$ = 1.001, $r_{rb}$ = 0.056).

As a follow-up, to check for object memory in each condition and group separately, in addition to our preregistered analyses, we conducted four Bayesian Wilcoxon signed rank tests testing whether the DLS was greater than 0. For mirror recognizers, there was strong evidence against a positive DLS for the other-owned objects ($BF_{10}$ = 0.085, W = 237, $\hat{R}$ = 1.003, $r_{rb}$ = −0.203) and inconclusive evidence for self-owned objects ($BF_{10}$ = 0.964, W = 361, $\hat{R}$ = 1.002, $r_{rb}$ = 0.213). For mirror non-recognizers, in contrast, there was evidence for a positive DLS for the other-owned objects ($BF_{10}$ = 8.197, W = 483, $\hat{R}$ = 1, $r_{rb}$ = 0.45) and evidence against a positive DLS for the self-owned objects ($BF_{10}$ = 0.236, W = 340, $\hat{R}$ = 1.003, $r_{rb}$ = 0.021). Notably, this means that non-recognizers only remembered the objects given to the puppet but not the objects assigned to themselves. In contrast, mirror recognizers tended to remember the self-assigned objects better than the objects given to the puppet, in line with the predicted self-reference effect.

In order to shed light on the potential mechanisms underlying enhanced memory for self-assigned objects in mirror self-recognizers and for other-assigned objects in non-recognizers, in addition to our preregistered analyses, we explored infants' looking times (LT) to the objects during the encoding phase. This revealed that, at encoding, mirror recognizers looked longer at the self-assigned objects (mean LT = 7.90 s, SD = 1.06 s) than the other-assigned objects (mean LT = 7.52 s, SD = 1.12 s, two-tailed Bayesian Wilcoxon signed rank test: $BF_{10}$ = 4.1, W = 427, $\hat{R}$ = 1.001, $r_{rb}$ = 0.435, N = 34). There was no such evidence for non-recognizers (self: mean LT = 7.8 s, SD = 1.09 s; other: mean LT = 7.58 s, SD = 0.89 s, two-tailed Bayesian Wilcoxon signed rank test: $BF_{10}$ = 0.4, W = 432, $\hat{R}$ = 1, $r_{rb}$ = 0.229, N = 37). Thus, in mirror recognizers, infants' enhanced attention to the self-assigned objects may have fostered their better encoding of these objects. In contrast, attention to the objects during encoding did not explain better encoding of other-assigned objects in non-recognizers.

Exploratorily, we also tested whether mirror-recognizers and non-recognizers differed with respect to other variables that may have contributed to the observed shift. We found no difference between the two groups in terms of their age (mean age of non-recognizers = 18.79 months, SD = 0.91, mean age of recognizers = 19.04 months, SD = 0.91, Bayesian Mann-Whitney U test: $BF_{10}$ = 0.509, W = 514.5, $\hat{R}$ = 1.001, $r_{rb}$ = −0.182) nor in terms of their inhibitory abilities (see SI p. 9, Supplementary Note 1), which we analyzed as a proxy for more general cognitive maturity[26].

We also did not find any effects of infants' use of verbal-self reference (i.e., their own name or first-person pronouns) on their memory of self- or other-assigned objects (Bayesian linear mixed model for the interaction between *ownership* and *use of own name*: $BF_{10}$ = 0.385, error = 6.121%, $\eta_p^2$ = 0.012; Main effect of own name: $BF_{10}$ = 0.308, error = 3.422%, $\eta_p^2$ = 0.012; Main effect of ownership: $BF_{10}$ = 0.259, error = 3.847%, $\eta_p^2$ = 0.001; Bayesian linear mixed model for the interaction between *ownership* and *pronoun use*: $BF_{10}$ = 0.999, error = 3.107%, $\eta_p^2$ = 0.037; Main effect of *pronoun use*: $BF_{10}$ = 0.258, error = 5.157%, $\eta_p^2$ = 0.006; Main effects of *ownership*: $BF_{10}$ = 0.269, error = 3.614%, $\eta_p^2$ = 0.004).

There was no evidence for any effects of sex (see SI, Supplementary Methods SM 5).

## Experiment 2: replication of familiarity preference for self-owned objects in mirror self-recognizers

Infants' memory may manifest as either a preference for the familiar or the novel stimulus[24]. Thus, in a second experiment, we sought to

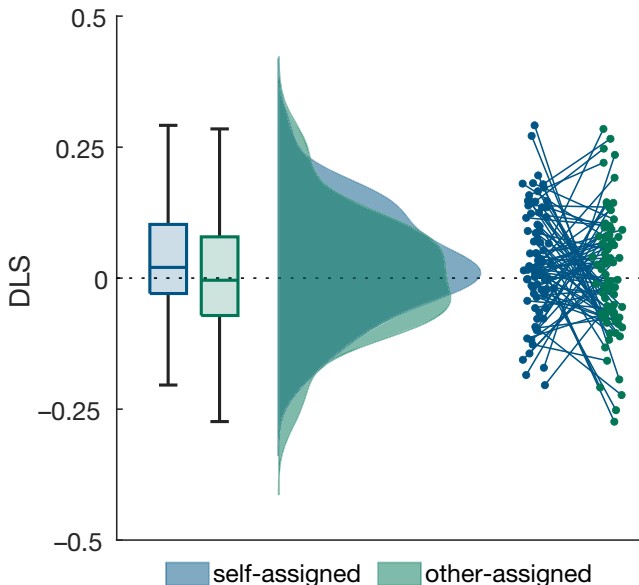

**Fig. 3 | Differential Looking Score (DLS) for self-owned versus other-owned objects in 20- to 40-month-old children (N = 82) in Experiment 2.** There was evidence for a familiarity preference in the self-assigned objects and evidence against a preference in the other-assigned objects. On each box, the central mark indicates the median, and the bottom and top edges of the box the 25th and 75th percentiles. The whiskers extend to the most extreme data points.

replicate the observed *familiarity* preference for the self-assigned objects in an older sample of toddlers. Further, in Experiment 1, mirror self-recognizers showed better memory (i.e., a greater DLS) for self-assigned than other-assigned objects, but they did not yet reliably remember the self-assigned objects (i.e., the average DLS for the self-owned objects was not significantly greater than 0). We reasoned that, if the self-reference effect emerges with the development of a self-concept, this effect may become more pronounced after 18 months of age, and memory for the self-owned objects should be better in slightly older children. In the second experiment, we therefore conducted an analogous version of the experiment in toddlers aged 20 to 40 months (N = 82, conducted online, as this follow up experiment was started during the pandemic in 2021). With this age range, we primarily sampled mirror self-recognizers, as also indicated via parental report due to the online format of the testing (details see "Methods").

Analogous to Experiment 1, we tested whether toddlers' DLS was positive for the self-owned but not the other-owned objects, to replicate the familiarity preference observed in Experiment 1 and infants' memory for the self-assigned objects. Indeed, this provided evidence for a positive DLS for the self-assigned objects (one-tailed Bayesian Wilcoxon signed rank test: $BF_{10} = 6.77$, W = 2215, $\hat{R} = 1.004$, $r_{rb} = 0.302$) and evidence against a positive DLS for the other-owned items ($BF_{10} = 0.20$, W = 1793, $\hat{R} = 1$, $r_{rb} = 0.054$, see Fig. 3).

For consistency with Experiment 1, we report the DLS instead of the preregistered raw LTs, but the preregistered analyses yield the same results with even stronger statistical evidence for a familiarity preference in the self-owned objects. Specifically, a directed Bayesian Wilcoxon signed rank test provided very strong evidence for longer looking times to familiar than unfamiliar self-assigned objects ($BF_{10} = 32.43$, W = 2295, $\hat{R} = 1.001$, $r_{rb} = 0.35$, mean $LT_{familiar} = 3.98$ s, $SD_{familiar} = 0.68$ s, mean $LT_{unfamiliar} = 3.72$ s, $SD_{unfamiliar} = 0.60$ s) and moderate evidence against longer looking times to familiar than unfamiliar other-assigned objects ($BF_{10} = 0.23$, W = 1828, $\hat{R} = 1$, $r_{rb} = 0.07$, mean $LT_{familiar} = 3.81$ s, $SD_{familiar} = 0.67$ s, mean $LT_{unfamiliar} = 3.75$ s, $SD_{unfamiliar} = 0.76$ s).

As in Experiment 1, exploratorily, we also tested whether, during the object encoding phase, infants' looking time to self-assigned objects (mean LT = 9.71 s, SD = 0.38 s) differed from their looking time to other-assigned objects (mean LT = 9.61 s, SD = 0.57 s) and found no evidence for a difference (two-tailed Bayesian Wilcoxon signed rank test remained inconclusive: $BF_{10} = 1.138$, W = 902, $\hat{R} = 1.001$, $r_{rb} = 0.31$).

Infants' familiarity preference for the self-owned objects did not change with age, indicating that the self-related memory benefit did not increase further between 20 and 40 months. Specifically, while children's total looking time increased with age ($BF_{10} = 193.8$; Kendall's tau B = 0.273, one-tailed), there was strong evidence against a positive correlation between age and the DLS for self-owned items ($BF_{10} = 0.077$, Kendall's tau B = −0.073, one-tailed, see Figure S3) as well as between age and the difference between the DLS for self-owned objects and the DLS for other-owned objects ($BF_{10} = 0.1$, Kendall's tau B = −0.039, one-tailed). When splitting the sample in two separate equal-sized age groups, similar results were obtained as for the entire age range (see SI, Supplementary Methods, SM 2.2). This indicates that the observed preference for self-relevant objects did not further increase with age but remained stable after 20 months. Therefore, we dropped age from our preregistered analyses. A Bayesian linear mixed model with age as a continuous predictor is reported in the SI (Supplementary Methods, SM 2.1).

As in experiment 1, there was no evidence for a relation of infants' memory for self- or other-assigned objects with their verbal self-reference (Bayesian linear mixed models for interaction between *ownership* and *use of own name*: $BF_{10} = 0.544$, error = 2.577%, $\eta_p^2 = 0.021$; main effect of *ownership*: $BF_{10} = 0.419$, error = 2.049%, $\eta_p^2 = 0.007$; main effect of *use of own name*: $BF_{10} = 0.232$, error = 5.738%, $\eta_p^2 = 0.003$; interaction between *ownership* and *pronoun use*: $BF_{10} = 0.415$, error = 4.296%, $\eta_p^2 = 0.082$; main effect of *ownership*: $BF_{10} = 0.418$, error = 1.231%, $\eta_p^2 = 0.016$; main effect of *pronoun use*: $BF_{10} = 0.062$, error = 8.773%, $\eta_p^2 = 0.006$). This lack of evidence for a relation with children's verbal self-reference is in line with the stability of the self-reference effect from 20 months of age. Indeed, verbal self-reference emerges later than mirror self-recognition between the second and third year of life. While the initial emergence of a self-concept (captured by mirror self-recognition) may be critical, verbal self-reference may more strongly be driven by linguistic developments.

No evidence for any effects of sex was observed (SI, Supplementary Methods, SM 5).

## Discussion

Our results demonstrate a shift in infants' memory with the emergence of mirror self-recognition around 18 months of age. Infants who recognized themselves in the mirror tended to remember objects better that had been assigned to themselves, in line with the well-known self-reference effect in adults. At the age of emergence of mirror self-recognition this effect was anecdotal, but we replicated this memory benefit for self-assigned objects (indexed by a familiarity preference) in an independent sample of slightly older toddlers who were mirror recognizers according to parental report. In contrast, infants who did not yet recognize themselves only remembered the objects that had been assigned to the puppet. The two groups of infants did not differ by age or general cognitive ability as indexed by inhibitory control, suggesting that infants' preferences were indeed driven by the development of their self-concept rather than by other more general indicators of maturity.

Mirror self-recognition is thought to reflect the emergence of a conceptual understanding of the self, as one is seen by others[13]. The shift towards better memory for self-related than for other-related content with the onset of mirror self-recognition thus suggests that the self-reference effect likely requires a conceptual representation of the self. Although our study does not reveal the factors that mediate this

relationship, our analyses showed that mirror self-recognizers spent more time attending to the self-assigned objects than the other-assigned objects during the encoding phase. This suggests, as others have also found, that self-related items may capture attention more, and that this may be a mechanism contributing to the enhanced memory for self-related items[5]. Our results suggest that a self-concept is necessary to encode an object as self-relevant. It has also been argued that the self-reference effect may arise from the ability to structure information in relation to the knowledge that we have of ourselves[6,8], which would again require a self-concept. In our paradigm, we used ownership (self-assigned vs. other-assigned) as a means of inducing self-relevance, reasoning that if an item can be understood as 'mine', it would acquire more relevance for the self. It has been proposed that superior memory for self-owned objects derives from imbuing those items with more value[27], leading to greater attention and encoding[4,5]. Whatever the reason for the relationship between ownership and the self-reference effect[5], we propose that, because mirror self-recognizers have a more developed self-concept, they were able to exploit this to construe the object placed in their own box in relation to the self. In addition, as infants were told that they will later be able to play with the objects placed in their box, this may have enhanced further the self-relevance of the objects by imbuing them with action-relevance.

In contrast, in a reversal of the self-reference effect, non-recognizers showed the opposite pattern. Not only did they show an absence of a self-related memory advantage, but they only remembered the objects assigned to the other. Although this finding was not predicted in the current study, it fits well with recent theorizing suggesting that infants may initially prioritize information relevant to others[11,28]. Young infants are strongly dependent on others, and, given their limited capacity to act on the world, it may initially be more relevant for them to allocate their attention to objects that afford action for others. As infants become increasingly capable of acting on the world themselves, and with the development of a self-concept and awareness of their capacity to act[12,29,30], the objects that they will be able to act on likely become more relevant to encode. Together with the increasing ability to reference these objects to the self, this may lead to the observed memory shift from an other- to a self-bias with the development of self-concept.

A similar bias for remembering objects that are relevant for others was recently observed in infants between 8 and 14 months, in the context of perspective taking[31–34]. Infants misremembered objects at the location where another person had seen them, or misremembered the contents of a box, even though infants themselves had seen the object move to a new location, or replaced with a new object[31–33]. This bias for prioritizing the perspective of others over their own also receded in the second year of life[31,32]. Such an altercentric bias was predicted under the framework of altercentrism, proposed to be a cognitive stance in which we encode the environment by considering the affordances that it offers for others in the vicinity. It has been suggested that, in infancy, a tendency to spontaneously encode the targets of others' attention is amplified because of the initial absence of a self-representation[11]. Although the utility of such an altercentric bias is necessarily speculative, it may facilitate infant learning by biasing attention to, and encoding of, the targets of others' attention and action. This would allow infants to prioritize learning about contents, which have already been curated by more knowledgeable others.

Importantly, our study suggests that this other-reference bias gives way to the classic memory bias for self-relevant content with the development of a conceptual self-representation, marking the emergence of the well-known self-reference effect. As the self-concept is emerging at around 18 months, infants may still experience a conflict between prioritizing self- and other-relevant content, which may have led to the poor overall object memory in the group of 18-month-old mirror self-recognizers. Indeed, while they tended to remember the

self-assigned objects better than the other-assigned objects, they did not remember the self-assigned objects above chance level. In turn, between 20 and 40 months, toddlers showed stable memory for the self-relevant objects but did not remember the other-relevant objects, and this self-related memory advantage did not further increase with age. A conflict between self- and other-biased memory around 18 months aligns with a recent study showing that the emergence of mirror self-recognition was related to increased conflict processing in a scenario where the infants' self-perspective was in conflict with that of another agent[35]. An exciting question for future research is how the balance of self- and other-bias may change over the preschool years, as this is a period of significant development of understanding others. A recent study indicates that more nuanced altercentric influences may re-emerge as children begin to understand conflicting perspectives at around 4 years of age[36]. Further, while an understanding of ownership develops early[23] and may therefore lead to the observed stable self-reference effect from early in toddlerhood, other forms of self-reference that rely on more complex self-knowledge (e.g., evaluating traits in relation to oneself) may only lead to memory benefits considerably later[37]. Accordingly, the self-reference effect for such contents may continue increasing with the development of self-knowledge through childhood.

While mirror self-recognizers attended longer to the self-assigned than the other-assigned objects during the object encoding phase, mirror non-recognizers did not show preferential attention to either of the objects at encoding. Yet, mirror non-recognizers later showed better memory of other-assigned than self-assigned objects. Thus, while attention to the object may provide a mechanism for better memory for self-assigned objects in the recognizers, it doesn't capture non-recognizers' enhanced memory for other-assigned objects. In previous work documenting a bias to remember the targets of others' attention in 8-month-old infants, we found that memory was greatest in infants who spent more time attending to the agent rather than the object[31], perhaps because altercentric encoding relies on attention between agent and object, whereas self-related encoding does not.

Could an increased understanding of ownership have contributed to the emerging memory benefit for self-assigned objects? While it seems likely that a more developed concept of self allows understanding when objects are assigned to the self (i.e., self-ownership), work shows that an abstract understanding of *others'* ownership is already present by 16 months[23], and is not associated with mirror self-recognition[22]. Further, an improved understanding of ownership cannot explain the observed transition from better memory for other-assigned to self-assigned objects in the current study. If younger infants had not yet understood the ownership manipulation, they should not have shown a memory benefit for the other-owned objects either. This indicates that both the mirror recognizers and the non-recognizers understood the ownership manipulation, and that non-recognizers prioritized remembering objects assigned to another agent while recognizers prioritized objects assigned to themselves. Nevertheless, it seems likely that a developing concept of self would allow the infant to appreciate self-ownership, which would presumably be necessary for an ownership manipulation to lead to the self-reference effect in memory.

In sum, our findings suggest that, before the emergence of a conceptual understanding of the self, infants have a memory bias for information relevant to others and prioritize this over content relevant to themselves. As the self-concept emerges in their second year, this altercentric bias recedes and transitions into a bias for self-relevant content, marking the onset of the self-reference effect well-documented in adult memory. We suggest that this initial bias for remembering what is relevant for others may be adaptive given young infants' strong dependency on others, which subsides as they become increasingly able to act independently in the second year of life.

## Methods

### Preregistration

Experiment 1 was preregistered at https://aspredicted.org/Z1H_H6F (10/23/2020), and Experiment 2 was preregistered at https://aspredicted.org/KIC_UFH (11/03/2021).

### Participants

**For Experiment 1**, N = 73 Danish-speaking children aged 17-21 months (median age = 18.6 months, 38 female) from the Copenhagen area were included in the analyses (the sample sizes resulted from a Bayesian Sequential testing scheme, see SI, Supplementary Methods, SM 3, Figure S4 and S5). Another 6 children were tested but excluded based on our preregistered criteria (see data preprocessing). The study was approved by the Ethics Committee of the Faculty of Social Sciences at Copenhagen University, parents provided informed consent for their children prior to participation, and children received a small gift.

**For Experiment 2**, N = 82 German-speaking children aged 20-40 months (median = 29 months, 49 female) were included in the analyses. Another 45 children participated in this experiment but had to be excluded because no data was recorded due to severe technical problems with the hosting platform Labvanced (N = 36), missing parental report on mirror self-recognition (N = 8), or not recognizing themselves in the mirror (N = 1, excluded as Experiment 2 aimed at replicating the familiarity preference for self-assigned objects in mirror self-recognizers). Children were recruited through the platform KinderSchaffenWissen.de and the database of the Max Planck Institute for Human Cognitive and Brain Sciences in Leipzig. The study was approved by the Ethics Committee of the Medical Faculty at the University of Leipzig and parents provided informed consent via button press. Both experiments complied with all relevant ethical regulations for the research with human participants.

### Procedures

**Experiment 1.** The self-reference task consisted of two phases—the encoding and the memory test phase. In the encoding phase, infants were seated opposite of the experimenter on their parent's lap at a table with two boxes, which were introduced as the infant's and a puppet Sven's box (see Fig. 1). To familiarize infants with the procedures and establish the relevance of the objects, 4 common objects were presented, assigned to either the infant or the puppet, and put in the respective boxes in alternating order. After this familiarization, infants were asked which box was theirs and were given the opportunity to play with the objects in their box for ~15 sec. During this time, the experimenter animated the puppet to play with his objects. In the object encoding phase, the experimenter then introduced a series of 16 handcrafted novel objects (see SI Figure S1), again introduced as either for the infant or for the puppet, in alternating order (objects and order counterbalanced across participants and randomly assigned). Each object was presented for ~10 seconds (either near the child's or near the puppet's box, but at an equal distance to the child, see Fig. 1) while saying "Look, this toy is for you/Sven. I will put it in your/Sven's box, and you/Sven can play with it later". The object was then put into the respective box outside of the child's view. There was no difference in the presentation duration of the self-assigned and other-assigned objects (two-tailed Bayesian Wilcoxon signed rank test: $BF_{10} = 0.138$, W = 1337, R = 1, $r_{rb} = -0.01$). For the memory test phase, infants were then seated on their parent's lap, at ~60 cm distance from a screen, and their gaze was recorded with an Eyelink1000 Plus eye-tracker. After a five-point calibration, infants were presented, on the screen, with photos of the previously introduced objects, next to a photo of a similar but modified object (see Fig. 1 and SI Figure S1). Each pair of objects was presented for 10 seconds and the infant's gaze toward each object was measured. Stimuli were presented with the software Experiment Builder.

**Experiment 2** was analogous to Experiment 1 but conducted online via the software Labvanced (as it started during the pandemic in 2021). At the beginning of the study, participants completed the experiment at home on their private laptop or PC (N = 57). This was done either without any experimenter or, due to substantial technical problems with the platform Labvanced, in a moderated setting, where an experimenter met with the parents remotely in a Jitsi video meeting, helped them set up the experiment, and stayed in the meeting in case further problems would occur. As many parents still experienced technical difficulties depending on their device, we finally invited children into our laboratory to conduct the Labvanced experiment on a lab computer (N = 25). In the lab, the experimenter set up the experiment for the parents and then left the room. Thus, the experimental procedure and setting in the lab were identical to the online experiment conducted from home, and no effect of setting on the results was found. Specifically, a mixed Bayesian ANOVA on the looking time to self-owned objects with *Novelty* as within factor and *Setting* (Online versus On Site) as between factor provided moderate evidence against a main effect of setting ($BF_{10} = 0.237$, error = 2.665%, $\eta_p^2 = 0.001$) and moderate evidence against an interaction ($BF_{10} = 0.221$, error = 3.543%, $\eta_p^2 < 0.001$).

Accordingly, in Experiment 2, both the object encoding phase and the memory test phase were presented on a screen. During the encoding phase, the two boxes were presented on the top and bottom of the right side of the screen, while the objects appeared, one by one, from the left side of the screen. An experimenter's voice introduced the objects as in experiment 1 and the objects then moved into the appropriate box (see: https://osf.io/n7p6j). Before the memory test phase, the infant's gaze was calibrated with the Labvanced infant calibration and both gaze and video data from the children was recorded with Labvanced. As the Labvanced webcam-based eye-tracking did not reliably track the infants' gaze, the direction of gaze was additionally coded manually from the videos using the software Boris 7.13. For the encoding phase, looking time at the screen was coded, for the memory test phase, gazes to the right and to the left side of the screen. As the webcam-based eye-tracking data was very noisy, and there was only a weak correlation between the average total looking times based on the webcam-based eye-tracking data and the manual coding (r = 0.336, BF = 14.83, see SI, Figure S2), we decided to move forward with the manual coding data (which had high interrater reliability). Twenty participants were coded by 2 independent coders with an interrater reliability (IRR) of >0.85 (two participants had lower IRR and a third rater worked as a tie-breaker).

### Data preprocessing and analysis

As preregistered, we excluded trials for which children did not look at the object for at least 5 out of 10 seconds during the object encoding phase and trials for which they did not fixate the screen for a total of at least 2 seconds during the memory test phase. According to standard practices, we also excluded trials with parental interference, experimenter, or technical errors. Children who did not provide at least 2 self-owned object and 2 other-owned object trials were excluded from analysis. To assess whether children looked longer to the familiar or the unfamiliar object, we calculated the differential looking score (DLS) as the fixation duration to the familiar minus the unfamiliar object divided by the total fixation duration to both objects. The sign of the DLS was inverted compared to the preregistered DLS (preregistered as fixation duration to the unfamiliar minus familiar) to allow for a more intuitive interpretation of the effects given the observed familiarity preference. Initially, we had defined the DLS the other way around based on a study investigating 11-month-old infants' memory of objects after manual exploration where infants showed a novelty preference[38]. Whether infants exhibit a familiarity or a novelty preference has been argued to depend on many factors, including age and task difficulty[24,25]. As the task by Begus et al.[38] was arguably easier

due to a longer exposure to the objects including manual exploration, it is not surprising that we observed a familiarity preference instead. In Experiment 1, fixations were defined as gaze points that did not move more than 50 pixels for at least 100 ms. As the DLS was not normally distributed (see SI, Supplementary Methods, SM 4, Figure S6 and S7), we used non-parametric alternatives to our preregistered statistical tests whenever possible. Wilcoxon signed rank tests were repeated with the sign test that does not assume symmetry, yielding highly comparable results (see SI, Supplementary Methods, SM 1). Statistical analyses were done with Matlab 2023b, JASP 0.18.3, R Studio 2024.04.0 and visualization with Matlab 2023b. Effect sizes were computed based on frequentist analyses in JASP 0.18.3.

## Mirror self-recognition test

In Experiment 1, infants additionally participated in a mirror self-recognition test (https://osf.io/2rw4k)[15,39]. In brief, infants were exposed to their mirror image, the mirror was then occluded, and a blue dot was surreptitiously applied to their nose. After 10 sec, the mirror was revealed again and the infants' behavior in front of the mirror observed. Mirror self-recognition was scored by whether the infant touched their face on or near the mark while looking at their mirror image (1 point) or not (0 points). In N = 29 infants, the mirror test was done after the self-reference task, and in N = 44 infants, it was done on a different day before the self-reference task (as these infants also participated in a different study where the mirror test had already been administered). The experimenter can thus be assumed to have been blind towards the mirror status of the infant. There was evidence against an effect of task order for other-owned objects (N = 44, mean DLS = 0.007, SD = 0.193; N = 29, mean DLS = −0.006, SD = 0.175; two-tailed Bayesian Mann-Whitney U test: $BF_{10}$ = 0.25, W = 644, $\hat{R}$ = 1.003, $r_{rb}$ = 0.009) and inconclusive evidence for self-owned objects (N = 44, mean DLS = 0.068, SD = 0.191; N = 29, mean DLS = −0.035, SD = 118; two-tailed Bayesian Mann-Whitney U test: $BF_{10}$ = 2.39, W = 428, $\hat{R}$ = 1.016, $r_{rb}$ = −0.329). In Experiment 2, mirror self-recognition was assessed via parent questionnaire, due to the online format of the study. Although asking parents whether their child recognizes themselves in the mirror introduced subjectivity to this measure, the children's age in Experiment 2 (20–40 months) made it safe to assume that, in the European context of the tested sample, the big majority of children would indeed recognize themselves in the mirror[16].

## Verbal self-reference use

Use of verbal self-reference (i.e., infants' own name and personal pronouns) was assessed via a parent questionnaire (see: https://osf.io/6et2h). Two scores were formed from the questionnaire data: First, whether they used their own name (1 point) or not (0 points) and second, whether they used any first-person pronouns (i.e., I, me, my/mine; 1 point) or not (0 points).

## Reporting summary

Further information on research design is available in the Nature Portfolio Reporting Summary linked to this article.

# Data availability

The anonymized coded behavioral data generated in this study have been deposited at OSF under https://doi.org/10.17605/OSF.io/Z3T8F[40]. The raw video data of the infants are protected and are not available due to data protection regulations. Further, several participants in Experiment 2 did not consent to data sharing, so that only a subset of this data is available.

# Code availability

Analysis code is available on OSF under https://doi.org/10.17605/OSF.io/Z3T8F[40].

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

## Acknowledgements

We would like to thank Dimitrios Askitis for his help with exporting eye-tracking data and quality control, and Kathrine Søndergaard Christensen, Josefine Bechgaard Lisse, Greta Dittmann, Jolien Beyer, Hannah Zielke, and Luna Taylor for their help with data acquisition. This research was funded by a Marie Słodowska-Curie Fellowship (ToMSelfLink 799734) and an ERC Starting Grant (REPRESENT 101117806) to CGW, an ERC Consolidator Grant (DEVOMIND 726114) to VS, and by the German Research Foundation (GR 5421/1-2) to KR.

## Author contributions

Conceptualization: C.G.W., E.H., V.S.; Methodology: C.G.W., K.R., E.H., E.Y.; Formal analysis: K.R., C.G.W., C.Y.; Investigation: K.H., E.H.; Data curation: K.H., E.H., C.Y.; Writing—original draft: C.G.W., K.R.; Writing—review & editing: C.G.W., K.R., C.Y., V.S.; Visualization: K.R., E.H., C.G.W.; Supervision: C.G.W., V.S.; Project administration: C.G.W., E.Y.; Funding acquisition: C.G.W., V.S.

## Funding

## Competing interests
The authors declare no competing interests.
