## [Transparent Peer Review file · Nature Communications]

The classic self-reference memory bias is preceded by an other-reference bias in infancy

Corresponding Author: Professor Charlotte Grosse Wiesmann

Version 0:

Reviewer comments:

Reviewer #1

(Remarks to the Author)

These results are noteworthy in two ways

1. They provide data on the earliest emergence of the self-reference effect - a robust memory bias for stimuli associated with the self-concept.
2. They provide evidence to support a fresh hypothesis in the developmental psychology literature - the proposal that infants are initially allocentric, and only move towards egocentrism with the onset of the self-concept.

The latter point elevates the importance of this work across various fields. The work is of significance not just for developmental psychologists, but for philosophers, anthropologists, neuroscientists and all disciplines interested in self, social cognition and memory. The boundaries drawn between self and other is a question of fundamental and enduring importance when considering the human condition.

The work is communicated clearly, and the literature review is thorough. The work is original, and robustly pre-registered, designed, executed, and analysed. The methods are described to a standard that can be easily reproduced and all materials are transparent. It is a significant achievement to redesign the self-reference paradigm to be age-appropriate for 18 month old infants, and the use of ownership - of notorious importance in this age range, is inspired.

I wholehearted support the publication of this important work and have only minor points to raise

1. It could be made a little clearer in the introduction WHY “information (the self-reference effect) may emerge in this period, in relation to infants’ emerging self concept”, explaining at this juncture in what way the self-concept might support memory
2. It could be clearer how mirror self-recognition was established for follow up sample 2 - were parents simply asked if children recognised in mirrors or provided information on how to test this at home?
3. I was a little unconvinced by the argument that changes infants understanding of ownership are not relevant to the switch – but only because I think a full understanding of ownership is inherently self-referent, and the literature on the (lack of) relation ownership and mirror self-recognition is limited. It seems theoretically intuitive that with the onset of the self-concept one’s understanding of ownership must necessarily be updated to include ‘my objects’, whereas prior to this there are only ‘other’s objects’. There may be objects I feel attached to/physically claim prior to the onset of the self-concept, but not objects I cognitively represent as ‘mine’.
4. Might the endowment effect -whereby owned objects are considered more valuable than non owned objects, even for ‘mere’ ownership effects (being told this is yours) - be relevant to the interpretation of your results? Like the self-reference effect, this effect is robust throughout the lifespan. I wondered if this might help to explain the switch from good memory for other objects - at the point where our understanding of ownership is defined as an object viewed as belonging to another person (see 3 above) – to bad memory for other’s objects, because I am now able to entertain the possibility of objects belonging to me – and this imbues them with value that other’s objects don’t have. Could your results be characterised as an endowment effect? Note: This is still inherently self-related.

Reviewer #2

(Remarks to the Author)

The present paper introduces a clever study on the emergence of self-reference bias as a function of the emerging self-concept measured by mirror-self recognition task. The results are claimed to support a developmental trend that an altercentric bias exists in the early months and with the emerging self concept, a novel focus of information organisation is appearing.

Overall, the study is well designed and interesting, the introduction is linear and coherent. The conclusions are mostly built on the data gathered. I have only some issues that could strengthen the paper.

1. In the literature review, the authors describe the early origins of the emergence of self concept, and link it to the mirror self-recognition task. While their focus is on a memory bias (self-reference bias), they do not cite a relevant literature that also highlights the role of self-concept in memory organisation. Howe and Courage's theory not only claims that the emerging self-concept could contribute the organisation of memories, but also could give an alternative explanation for the emergence of the self-reference bias that the authors should consider.

Howe, M. L., & Courage, M. L. (1997). The emergence and early development of autobiographical memory. *Psychological Review*, 104(3), 499–523. <https://doi.org/10.1037/0033-295X.104.3.499>

2. The authors claim: „As memory in infants may manifest as either a familiarity or a novelty preference 24” p.4, lines 103-104; and they do not predict the direction of preference, just the preference per se. Indeed, in the preregistration they introduced the DLS score as 'fixation duration to the unfamiliar minus familiar', suggesting that the author's intuition was that children would rather show novelty preference. I suggest the authors to describe in more detail the variability of the direction of preference, and their intuitive approach.

3. The analyses focus on the DLS scores, and as follow-up analyses, Wilcoxon signed rank tests were performed. I see that the the Wilcoxon signed-rank makes use of the magnitudes of the differences rather than just their signs, and therefore it could be more robust, however, the signed rank test carries an assumption about symmetry of differences under the null that the sign test don't. Given that the authors do not predict the direction of preference beforehand, and relatedly, could not necessarily assume such symmetry, it would be very important to run a sign test for the relevant comparisons as well.

4. I found the discussion somewhat unbalanced. While the paper is supposedly about the emergence of self-reference bias, the discussion describes and tries to explain the opposite pattern, the other-reference bias as a dominant pattern preceding the self-reference bias. While this is a very interesting possibility, the data supporting this conclusion is coming from only Experiment 1 (this set of results is not replicated in Exp 2). I suggest the authors to handle this finding with more reservation.

I think the paper would benefit from considering and adding information on the above points to the ms.

Reviewer #3

(Remarks to the Author)

The present manuscript investigates the self-reference effect in infants' and children's memory, demonstrating that differences in memory for familiar stimuli differ as a function of agent (self vs. other) and mirror self-recognition performance (pass or fail; as a proxy for children's self concept). The authors argue that the self-reference effect emerges coincident with self-recognition, and perhaps more surprisingly, that prior to successful self-recognition, infants actually show an other-reference effect, supporting the authors' prior arguments that infants, before developing a sense of self, possess an altercentric stance.

As a starting point, I am a big fan of the recent move by this lab group to test the long-held idea that development consists of movement away from an egocentric perspective; the general idea that infants may in fact be **altercentric** early on is novel and exciting and, if correct, will lead to a paradigm shift in our understanding of infants and will turn the interpretation of many research findings on their respective heads. This paper represents a systematic advance of this perspective, and provides a critical experimental test. This is all to say that this is a very exciting topic and the results are critically important to developmental psychology, amongst other fields.

I found the paper to be well written, and easy to follow. Primarily my concerns surround critical aspect of the methods and coding. I detail these concerns below.

1. I found the approach of introducing objects as belonging to the self vs. other, then tracking infants' memory for these objects vs. a novel distractor via eye gaze to be elegant and simple. However, taking this approach to answer the question at hand seems problematic in two ways. First, there seems to be something a bit circular here: if infants that fail self recognition tests don't have a sense of self, then how could they possibly encode something in a self-referential way? Put differently, doesn't the core hypothesis preclude the use of this particular paradigm? Second, and setting my first concern aside, what evidence do we have that infants actually understand some toys as self assigned and other toys as other assigned? It seems like some sort of manipulation check is needed here.

One could argue that the results in and of themselves provide evidence that infants differentially associate toys with self and other. But here is the rub: as the authors acknowledge in the introduction, infants sometimes show familiarity preferences

and other times novelty preferences. Let's say, for the sake of argument, that non-mirror recognizers show a familiarity preference (as the manuscript assumes) but mirror-recognizers show a novelty preference (opposite the interpretation in the manuscript). In this case the results would be rather commensurate across groups (i.e., both groups would appear altercentric). While this interpretation may seem like a stretch, the two groups (recognizers and non-recognizers) may differ in age (I didn't see a statistical comparison of the recognizers and non-recognizers in terms of age), general cognitive development etc, that is in turn linked to novelty versus familiarity preferences.

Experiment 2 helps a little with this issue - it shows that older kids reliably show a familiarity preference - but doesn't resolve it fully as we know that familiarity versus novelty preferences can differ by age, information processing abilities, attention during encoding (see below), etc.

What would help is a separate experiment in which the same objects are presented "agentless" with 18 month olds (i.e. same procedure with a box unrelated to an agent during encoding). Again, a mirror self-recognition task could be administered to split the group into recognizers versus non-recognizers; the researchers here should find that regardless of MSR status infants show a familiarity preference. This experiment would be a lynchpin for the authors' interpretation.

2. Related to above, can the authors demonstrate that recognizers versus non-recognizers did not differ based on MSR performance by age, information processing abilities, and other measures of general developmental status?

3. It is important to ensure that infants looked equally long to self and other toys during encoding (both as a group and for each sub-group based on infants' mirror self recognition performance); otherwise it is possible that test performance simply reflects how attentive infants were to various objects during encoding. Of course, if there are differences in encoding as a function of agent and mirror self-recognition that may be interesting in and of itself, but it would be potentially a different effect than the exact effect the authors argue for here. I get the impression that this data is at least partially available based on the exclusion criteria in the detailed methods; it should be included and analyzed in the main text.

3. Related to the above, and also an independent concern: to what degree was coding done of the experimental procedure to ensure that self vs. other items were introduced in the same manner? It is important to undertake some coding to this end because obviously the experimenter could not be blind to which toys were the infants' versus puppets' toys. Similarly, was the experimenter aware of infants' self-recognition status?

4. In the discussion, I'd love to see more about the potential utility of the altercentric bias. I find the current writing to be a bit hand-wavy.

5. Throughout there are places in which the wording implies that infants' memory for self-associated objects versus other associated objects were directly compared against one another (i.e., on test, shown self versus other objects and their preference recorded). I'd suggest clarifying throughout the manuscript. Relatedly, this wording made me wonder why the authors didn't take this approach.

In summary, I think this manuscript is an interesting attempt to directly test the altercentric stance perspective, and tie the erosion of this stance to the emergence of a sense of self. However, more work needs to be done to convince the reader that the findings reflect what the authors believe they reflect. While my comments may seem critical, I'd like to underscore that I raise these concerns, and hope the authors will address them, because I think that if the initial altercentric stance theory is correct it will revolutionize the field! For these reasons, I want to ensure that this novel and important theory is tested in the most rigorous of ways.

Version 1:

Reviewer comments:

Reviewer #1

(Remarks to the Author)

This revised manuscript is responsive to all of the issues raised by reviewers and provides convincing rebuttals/clarifications to all points. I very much look forward to seeing this excellent article in print.

Reviewer #2

(Remarks to the Author)

The revised version of this paper is much improved. I appreciate the authors' careful handling of my previous comments. In details, the theoretical framework is now adequately elaborated, the additional analysis revealed convergent results, and also the discussion of the findings is more clearly delivers the main message.

Reviewer #3

(Remarks to the Author)

I think the authors have done a thorough and admirable job in attending to my prior comments and concerns. I endorse this

manuscript for publication.

RESPONSE TO THE REVIEWER COMMENTS

Reviewer #1:

These results are noteworthy in two ways

1. They provide data on the earliest emergence of the self-reference effect - a robust memory bias for stimuli associated with the self-concept.
2. They provide evidence to support a fresh hypothesis in the developmental psychology literature - the proposal that infants are initially allocentric, and only move towards egocentrism with the onset of the self-concept.

The latter point elevates the importance of this work across various fields. The work is of significance not just for developmental psychologists, but for philosophers, anthropologists, neuroscientists and all disciplines interested in self, social cognition and memory. The boundaries drawn between self and other is a question of fundamental and enduring importance when considering the human condition. The work is communicated clearly, and the literature review is thorough. The work is original, and robustly pre-registered, designed, executed, and analysed. The methods are described to a standard that can be easily reproduced and all materials are transparent. It is a significant achievement to redesign the self-reference paradigm to be age-appropriate for 18 month old infants, and the use of ownership - of notorious importance in this age range, is inspired.

I wholeheartedly support the publication of this important work and have only minor points to raise.

We thank the reviewer for their positive evaluation of our study and for the helpful comments.

1. It could be made a little clearer in the introduction WHY “information (the self-reference effect) may emerge in this period, in relation to infants’ emerging self-concept”, explaining at this juncture in what way the self-concept might support memory

We have now elaborated more on why the emerging self-concept might support memory in the first paragraph of the Introduction (**p. 3, line 43**), and in the suggested paragraph:

Introduction, p. 4, line 74:

“In the second year of life, infants undergo important developments in their self-awareness that we reasoned may contribute to the development of the self-reference effect¹¹, either because of increased attention to items that may now be encoded as self-relevant, or because of differences in organization of incoming information in relation to the emerging self-concept.^{8”}

2. It could be clearer how mirror self-recognition was established for follow up sample 2 - were parents simply asked if children recognised in mirrors or provided information on how to test this at home?

Parents were simply asked whether children recognize themselves in the mirror. We have now clarified this in the manuscript and pointed this out as a potential limitation (as pasted below). Given that the children in sample 2 were 20 to 40 months old, however, it is safe to assume that the big majority of them would indeed pass the mirror test - based on the extensive literature on mirror self-recognition showing that, in Western populations, more than 80% of infants recognize themselves in the mirror by 20 months and 100% by 24 months (e.g., Lewis & Ramsay, 2004).

Methods, p. 14, line 359:

" In Experiment 2, mirror self-recognition was assessed via parent questionnaire, due to the online format of the study. Although asking parents whether their child recognizes themselves in the mirror introduced subjectivity to this measure, the children's age in experiment 2 (20 to 40 months) made it safe to assume that, in the European context of the tested sample, the big majority of children would indeed recognize themselves in the mirror¹⁶."

3. I was a little unconvinced by the argument that changes in infants' understanding of ownership are not relevant to the switch – but only because I think a full understanding of ownership is inherently self-referent, and the literature on the (lack of) relation [between] ownership and mirror self-recognition is limited. It seems theoretically intuitive that with the onset of the self-concept one's understanding of ownership must necessarily be updated to include 'my objects', whereas prior to this there are only 'other's objects'. There may be objects I feel attached to/physically claim prior to the onset of the self-concept, but not objects I cognitively represent as 'mine'.

We agree with the reviewer that as one's concept of self develops, so does one's ability to understand objects as 'belonging to the self'. What we were trying to argue is that an understanding of ownership itself (independently of the self) is unlikely to emerge from an understanding of self. This is because a) there is evidence that younger infants (16-month-olds) have some degree of understanding of ownership when it comes to *others' relationship with objects* (Stahl et al. 2023) and b) in our study, infants who were non-recognizers appeared to understand the ownership element of the task because they showed a greater memory for objects assigned to the puppet. So, we argue that it cannot be that the non-recognizers required a concept of self to understand the ownership manipulation – because they understood when objects were assigned to the puppet (because they remembered these objects better than the

objects assigned to themselves). We agree that understanding objects as belonging to the self must require a concept of self and have now acknowledged this in the discussion:

Discussion, p. 12, line 302 and 311:

“While it seems likely that a more developed concept of self allows understanding when objects are assigned to the self (i.e., self-ownership), work shows that an abstract understanding of others’ ownership is already present by 16 months²⁴, and is not associated with mirror self-recognition²³. (...)”

“Nevertheless, it seems likely that a developing concept of self would allow the infant to appreciate self-ownership, which would presumably be necessary for an ownership manipulation to lead to the self-reference effect in memory.”

4. Might the endowment effect - whereby owned objects are considered more valuable than non owned objects, even for 'mere' ownership effects (being told this is yours) - be relevant to the interpretation of your results? Like the self-reference effect, this effect is robust throughout the lifespan. I wondered if this might help to explain the switch from good memory for other objects - at the point where our understanding of ownership is defined as an object viewed as belonging to another person (see 3 above) – to bad memory for other’s objects, because I am now able to entertain the possibility of objects belonging to me – and this imbues them with value that other’s objects don’t have. Could your results be characterised as an endowment effect? Note: This is still inherently self-related.

Thank you for this suggestion. Indeed, it has been proposed that the endowment effect is predicated on the self-referential encoding biases that are associated with ownership, because pairing an item with the self is sufficient to enhance its perceived value (Beggan, 1992). We have now extended the Discussion to include a more in-depth discussion of what might underlie the superior memory for self-assigned objects in mirror-recognizers, including the possibility that understanding objects as self-owned imbues them with more value.

p. 10 line 223:

“In our paradigm, we used ownership (self-assigned vs. other-assigned) as a means of inducing self-relevance, reasoning that if an item can be understood as ‘mine’, it would acquire more relevance for the self. It has been proposed that superior memory for self-owned objects derives from imbuing those items with more value²⁸, leading to greater attention and encoding^{4,5}.”

Reviewer #2:

The present paper introduces a clever study on the emergence of self-reference bias as a function of the emerging self-concept measured by mirror-self recognition task. The results are claimed to support a developmental trend that an altercentric bias exists in the early months and with the emerging self concept, a novel focus of information organisation is appearing.

Overall, the study is well designed and interesting, the introduction is linear and coherent. The conclusions are mostly built on the data gathered. I have only some issues that could strengthen the paper.

We thank the reviewer for their positive evaluation of our study and their valuable suggestions, which we have addressed as explained point-by-point below.

1. In the literature review, the authors describe the early origins of the emergence of self concept, and link it to the mirror self-recognition task. While their focus is on a memory bias (self-reference bias), they do not cite a relevant literature that also highlights the role of self-concept in memory organisation. Howe and Courage's theory not only claims that the emerging self-concept could contribute to the organisation of memories, but also could give an alternative explanation for the emergence of the self-reference bias that the authors should consider.

Howe, M. L., & Courage, M. L. (1997). The emergence and early development of autobiographical memory. *Psychological Review*, 104(3), 499–523. <https://doi.org/10.1037/0033-295X.104.3.499>

We have now incorporated the reviewer's suggestion. As we understand it, a dominant explanation of the self-reference bias in memory is due to how the self-concept organizes memory. In the Introduction, we write:

p. 3, line 46:

"Beyond attention allocation, there are several other theoretical explanations for why self-relevant information is remembered better, including the possibility for more elaborative encoding and better organization of items within the framework of rich conceptual self-knowledge⁶⁻⁸."

p. 4, line 74:

"In the second year of life, infants undergo important developments in their self-awareness that we reasoned may contribute to the development of the self-reference effect¹¹, either because of increased attention to items that may now be encoded as self-relevant, or because of differences in organization of incoming information in relation to the emerging self-concept⁸."

Reference 8 refers to Howe & Courage (1997).

2. The authors claim: „As memory in infants may manifest as either a familiarity or a novelty preference 24” p.4, lines 103-104; and they do not predict the direction of preference, just the preference per se. Indeed, in the preregistration they introduced the DLS score as ‘fixation duration to the unfamiliar minus familiar’, suggesting that the author’s intuition was that children would rather show novelty preference. I suggest the authors to describe in more detail the variability of the direction of preference, and their intuitive approach.

Our intuition of a novelty preference was based on a previous study of one of the authors with younger infants measuring memory for objects (Begus, Southgate & Gliga, 2015). In that study, infants showed a novelty preference when presented with a similar looking-time test, but prior to this had the opportunity to manually explore the objects for as long as they liked. In the current study, infants did not manually explore the objects and only saw them for 10 seconds maximum. Infants in both experiment 1 and experiment 2 showed a familiarity preference (i.e., they looked longer at the object they had previously seen). Whether infants manifest memory as a novelty or familiarity preference depends on various factors, including age and task difficulty (Hunter & Ames, 1988). We therefore assume that encoding the objects in our task was made more difficult by infants not having the opportunity to manually explore the objects and the relatively short exposure to the objects, and that this is why infants’ memory was reflected in a familiarity rather than a novelty preference. We have now added this reasoning to the manuscript.

p. 15, line 383:

*“The sign of the DLS was inverted compared to the preregistered DLS (preregistered as fixation duration to the unfamiliar minus familiar) to allow for a more intuitive interpretation of the effects given the observed familiarity preference. **Initially, we had defined the DLS the other way around based on a study investigating 11-month-old infants’ memory of objects after manual exploration where infants showed a novelty preference³⁹. Whether infants exhibit a familiarity or a novelty preference has been argued to depend on many factors, including age and task difficulty^{25,26}. As the task by Begus et al.³⁹ was arguably easier due to a longer exposure to the objects including manual exploration, it is not surprising that we observed a familiarity preference instead.**”*

3. The analyses focus on the DLS scores, and as follow-up analyses, Wilcoxon signed rank tests were performed. I see that the Wilcoxon signed-rank makes use of the magnitudes of the differences rather than just their signs, and therefore it could be more robust, however, the signed rank test carries an assumption about symmetry of differences under the null that the sign test don’t. Given that the authors do not predict the direction of preference beforehand, and relatedly, could not necessarily

assume such symmetry, it would be very important to run a sign test for the relevant comparisons as well.

We thank the reviewer for pointing us to this issue. We have now repeated all analyses with the sign test instead of the Wilcoxon signed-rank test, which yields highly similar results (as reported in the **Supporting Information, p. 3, section S4**). Since the Wilcoxon signed-rank test is more robust and powerful (due to taking the magnitudes of the differences into account), we report the Wilcoxon signed-rank test in the main manuscript.

4. I found the discussion somewhat unbalanced. While the paper is supposedly about the emergence of self-reference bias, the discussion describes and tries to explain the opposite pattern, the other-reference bias as a dominant pattern preceding the self-reference bias. While this is a very interesting possibility, the data supporting this conclusion is coming from only Experiment 1 (this set of results is not replicated in Exp 2). I suggest the authors to handle this finding with more reservation.

We understand the reviewer's comment, and we have attempted to redress this balance by extending our discussion of the self-reference effect. In particular, we have now provided more discussion on factors that may give rise to the emergence of the self-reference effect in relation to the self-concept in early development (**p. 10, line 227**).

In addition, we now point out that, although we didn't predict the prioritization of other-assigned objects in the non-recognizers in the current study, this finding aligns well with an increasing body of evidence for an altercentric bias in young infants (Manea et al., 2023; Kampis et al., 2024; Tebbe et al. 2024a, 2024b) and the theoretical prediction of its receding with the development of self-concept (Southgate, 2020; Grosse Wiesmann & Southgate 2021). We think it is exciting that we turned up such a bias in the non-recognizers in a different context related to ownership, and that it indeed turns around with mirror self-recognition as predicted by the theory. We have now explicitly noted in the Discussion that this was not a predicted effect:

p. 10, line 246:

"Although this finding was not predicted in the current study, it fits well with recent theorizing suggesting that infants may initially prioritize information relevant to others^{11,29}."

We did not test this effect in Exp. 2 as Exp. 2 was with older infants (20-40 months) who were all mirror recognizers. Thus, we would not expect to find an 'other-reference' effect in this older group of infants.

Reviewer #3:

The present manuscript investigates the self-reference effect in infants' and children's memory, demonstrating that differences in memory for familiar stimuli differ as a function of agent (self vs. other) and mirror self-recognition performance (pass or fail; as a proxy for children's self concept). The authors argue that the self-reference effect emerges coincident with self-recognition, and perhaps more surprisingly, that prior to successful self-recognition, infants actually show an other-reference effect, supporting the authors' prior arguments that infants, before developing a sense of self, possess an altercentric stance.

As a starting point, I am a big fan of the recent move by this lab group to test the long-held idea that development consists of movement away from an egocentric perspective; the general idea that infants may in fact be **altercentric** early on is novel and exciting and, if correct, will lead to a paradigm shift in our understanding of infants and will turn the interpretation of many research findings on their respective heads. This paper represents a systematic advance of this perspective, and provides a critical experimental test. This is all to say that this is a very exciting topic and the results are critically important to developmental psychology, amongst other fields.

I found the paper to be well written, and easy to follow. Primarily my concerns surround critical aspects of the methods and coding. I detail these concerns below.

We thank the reviewer for their positive evaluation of our study and their excellent comments, which we have addressed as explained in detail below.

1. I found the approach of introducing objects as belonging to the self vs. other, then tracking infants' memory for these objects vs. a novel distractor via eye gaze to be elegant and simple. However, taking this approach to answer the question at hand seems problematic in two ways. First, there seems to be something a bit circular here: if infants that fail self recognition tests don't have a sense of self, then how could they possibly encode something in a self-referential way? Put differently, doesn't the core hypothesis preclude the use of this particular paradigm?

We thank the reviewer for pointing out this unclarity in the premises and hypotheses of our study. To clarify that there is no circularity, we now explain that there are different mechanisms that have been postulated to underlie the self-reference effect, which make distinct predictions as to when infants should show enhanced memory for self-assigned objects in our paradigm. One possibility is that the richness of conceptual self-knowledge helps with a better organization of memory in relation to this body of knowledge. This would clearly require a self-concept, although it remains an open question how elaborate self-knowledge needs to be, and whether infants would already show a self-reference effect when the concept of self first emerges, or

only later as self-knowledge becomes more elaborate. Another possibility is that the self-reference effect results from better memory for information that is action- or goal-relevant. Such a mechanism could well already be in place before the emergence of a self-concept. Thus, in our paradigm, infants would not need to understand the manipulation as relevant to the *self*, or *self*-referential, but instead could simply process the toys assigned to them as more relevant because they will be able to play with them (i.e., as relevant to their goal of playing with toys). It is therefore not the case that, in our paradigm, infants necessarily have to encode the objects as self-referential in order to show enhanced encoding of the self-assigned objects (e.g., they could have shown enhanced encoding due to their action-relevance before the emergence of a self-concept). Instead, this is precisely our hypothesis that infants begin to show enhanced encoding of self-assigned objects as soon as they understand them as referring to the self, and that they do this as the first robust indicators of self-concept emerge, namely as they recognize themselves in the mirror (and not only later with more elaborate self-knowledge). In sum, different theories make different predictions on when the self-reference effect emerges and when enhanced memory for the self-assigned objects shows in our paradigm, showing that it is not circular.

We have now added a discussion of these different possible mechanisms to the first paragraph of the Introduction (**p.3, line 43**) and to the Discussion (**p.10, line 227**), and we hope that by illustrating the different possible predictions for our paradigm, we have clarified why there is no circularity in our paradigm or research question.

In addition, we now explain in more depth that an understanding of ownership (that is manipulated in this paradigm) does not require a self-concept. Other studies have shown that an understanding of ownership exists by 16 months and is not related to mirror self-recognition (**p.12, line 302**). In our paradigm, Infants without mirror self-recognition apparently understood the 'ownership' manipulation because they showed a memory bias for the objects assigned to the other. If they had not understood the manipulation, they should not have shown a difference between self- and other-assigned objects.

Second, and setting my first concern aside, what evidence do we have that infants actually understand some toys as self assigned and other toys as other assigned? It seems like some sort of manipulation check is needed here.

As the reviewer mentions in their next comment, our pattern of results implies that the infants understood the manipulation, assuming infants indeed showed a familiarity preference, which we have strong evidence for, as explained in response to the next comment. If they had not understood the manipulation, they should not have shown a difference between self- and other-

assigned objects. Instead, mirror recognizers showed better memory for self-assigned than other-assigned objects in line with our main preregistered prediction. This provides evidence that they understood the self-assignment and differentiated it from the other-assignment. Non-recognizers showed better memory for other-assigned objects, indicating that they understood the assignment to the other and differentiated it from the self-assignment.

One could argue that the results in and of themselves provide evidence that infants differentially associate toys with self and other. But here is the rub: as the authors acknowledge in the introduction, infants sometimes show familiarity preferences and other times novelty preferences. Let's say, for the sake of argument, that non-mirror recognizers show a familiarity preference (as the manuscript assumes) but mirror-recognizers show a novelty preference (opposite the interpretation in the manuscript). In this case the results would be rather commensurate across groups (i.e., both groups would appear altercentric). While this interpretation may seem like a stretch, the two groups (recognizers and non-recognizers) may differ in age (I didn't see a statistical comparison of the recognizers and non-recognizers in terms of age), general cognitive development etc, that is in turn linked to novelty versus familiarity preferences. Experiment 2 helps a little with this issue - it shows that older kids reliably show a familiarity preference - but doesn't resolve it fully as we know that familiarity versus novelty preferences can differ by age, information processing abilities, attention during encoding (see below), etc.

If we understand correctly, the reviewer suggests that the two groups might differ in the direction in which they manifest their discrimination because of differences in age or cognitive maturity. Thus, the reviewer argues that, for mirror-recognizers, what we interpret as a stronger familiarity preference (i.e., a more positive DLS) in the self-assigned condition, could possibly be a novelty preference (i.e., a more negative DLS) for the other-assigned condition. And if so, then they would look more like the non-recognizers who show a familiarity preference for other-assigned objects. We would like to offer 4 points that speak against this possibility:

- A) As the reviewer points out, age is indeed a factor that influences whether infants manifest a novelty or familiarity preference. We have therefore now checked this potential concern and show that the groups are the same age (mean age of recognizers = 18.79 months, SD = 0.91, mean age of non-recognizers = 19.04 months, SD = 0.91, Mann-Whitney U test: $BF=0.509$, $W=514.5$, $R=1.001$). We have now reported this data in the manuscript (**Results, p. 7 line 161**).
- B) The reviewer also points out that the mirror recognizers may be more mature than the non-recognizers and that this could influence whether they exhibit a novelty or a familiarity preference. As $N = 44$ of the infants of experiment 1 also participated in a different study, we had access to their performance on the Early Childhood Inhibition Touch Screen Task

(ECITT, Holmboe et al. 2021), which is a measure of inhibitory control. Following the reviewer's suggestion, we now included this data to check that there were not more broad differences in cognitive maturity between the mirror recognizers and non-recognizers. This confirms that there are no differences in performance on this task (see **Results, p. 7 line 165**; and **Supporting Information, p. 7 section S8**). We thank the reviewer for helping us to strengthen our conclusions with this data, which we have now included in the manuscript along with a discussion of its relevance (**Discussion, p. 9 line 210**).

- C) While there is statistical evidence for a familiarity preference ($DLS > 0$) both in the non-recognizers in experiment 1 (for other-assigned objects) and in the recognizers in experiment 2 (for self-assigned objects), at no point in the data do we observe a negative DLS that would support a novelty preference. Indeed, for the mirror-recognizers, there is strong evidence that the DLS does not differ from 0 for the other-assigned objects ($BF = 0.085$). If the reviewer's interpretation was correct, this DLS would have to be negative.
- D) Finally, if the reviewer's concern was right, it would mean that infants would need to shift from a familiarity preference before they achieve mirror self-recognition, to a novelty preference as they achieve mirror self-recognition, back to a familiarity preference again shortly after (that we observe in the infants 20-40 months in experiment 2). We are not aware of any theory that would predict such a back-and-forth shift in preferences, particularly not within such a short period of only 2 months. We therefore think that it is a lot more plausible to assume that, in this task, in the entire tested age range, infants reveal their memory through a familiarity preference.

We hope that the data showing a) no age differences between the two groups b) no differences in a standard inhibitory control task, often interpreted as one of cognitive maturity, between the two groups, c) no statistical evidence for a novelty preference in any of our conditions or groups, and d) an implausible shift from familiarity to novelty back to familiarity preference within a very short period that would have to be true for the reviewer's concern to be borne out, will alleviate the reviewer's concerns.

What would help is a separate experiment in which the same objects are presented "agentless" with 18 month olds (i.e. same procedure with a box unrelated to an agent during encoding). Again, a mirror self-recognition task could be administered to split the group into recognizers versus non-recognizers; the researchers here should find that regardless of MSR status infants show a familiarity preference. This experiment would be a lynchpin for the authors' interpretation.

We hope that based on our new analyses and arguments above, we have convinced the reviewer that there is no basis for assuming anything other than a familiarity preference. Furthermore, while the suggested agentless experiment would be interesting in its own right, we think that it would not be able to clarify whether infants show a familiarity or novelty preference in our agent version of the experiment. This is because whether infants show a familiarity or a novelty preference is likely based on many factors, not only age and cognitive maturity, but also task complexity/difficulty (Hunter & Ames, 1988). Removing the agent would clearly change the complexity and difficulty of the task. It would likely reduce complexity because the infant would not need to consider the agent factor or encode the relationship between the box and the other. This could lead to a novelty, rather than familiarity effect. It may also increase difficulty as there would be no agent to enhance memory, potentially leading to no preference at all. Thus, no matter whether the results would reveal a novelty preference, a familiarity preference, or no preference at all, it is not clear what we could conclude from this for our agent version of the experiment. As we already have evidence for a familiarity preference across 3 different groups, under identical circumstances, we believe this is good evidence that this paradigm does not elicit a novelty effect in any of our groups.

2. Related to above, can the authors demonstrate that recognizers versus non-recognizers did not differ based on MSR performance by age, information processing abilities, and other measures of general developmental status?

As mentioned above, we now checked whether mirror recognizers and non-recognizers differed in terms of age and inhibitory abilities (as an index of more general cognitive maturity), and we found statistical evidence against a difference between the two groups (see Results section **p. 7 line 161**, and **Supporting Information: p. 7, section S8**). We have now also added these findings to the abstract and we discuss their relevance in the Discussion (as pasted below). We would like to thank the reviewer for helping us to strengthen our conclusions with this data.

Discussion, p. 9 line 217:

The two groups of infants did not differ by age or general cognitive ability as indexed by inhibitory control, suggesting that infants' preferences were indeed driven by the development of their self-concept rather than by other more general indicators of maturity.

3. It is important to ensure that infants looked equally long to self and other toys during encoding (both as a group and for each sub-group based on infants' mirror self recognition performance); otherwise it is possible that test performance simply reflects how attentive infants were to various objects during encoding. Of course, if there are differences in encoding as a function of agent and mirror self-recognition that may be interesting in and of itself, but it would be potentially a different effect than the

exact effect the authors argue for here. I get the impression that this data is at least partially available based on the exclusion criteria in the detailed methods; it should be included and analyzed in the main text.

We thank the reviewer for this valuable comment and have now checked this. Interestingly, the mirror self-recognizing infants do look longer towards the self-assigned objects during the encoding phase than the other-assigned objects. The mirror non-recognizers, in turn, do not show a difference in attention at encoding. We now report these data in the Results section (**p. 6 line 150**). We agree with the reviewer that this is an interesting observation in itself, which may shed some light on the mechanisms underlying our effect. Indeed, we interpret this as further support for our hypothesis that, as the self-concept emerges, infants begin to show enhanced attention to self-relevant information, fostering their encoding of this information. It is well established that self-relevant information captures and holds attention (e.g. Turk et al., 2011, Sui et al., 2015) and that it is this increase in attentional resources that is hypothesized to facilitate the encoding of such stimuli, leading to the self-reference effect in memory (Turk et al., 2008; Cunningham et al., 2014). Thus, we do not consider the greater attention during the encoding phase to suggest a different effect; rather, we assume that greater attention during the encoding phase might be a mechanism that supports enhanced encoding of self-relevant items. However, while this is the case for the self-assigned objects in mirror recognizers, attentional differences do not explain the better memory for other-assigned objects in the non-recognizers. Namely, infants who do not show mirror self-recognition did not evidence differential attention to either object. We have now addressed this point in the Discussion.

Discussion, p. 10, line 230:

“Although our study does not reveal the factors that mediate this relationship, our analyses showed that mirror self-recognizers spent more time attending to the self-assigned objects than the other-assigned objects during the encoding phase. This suggests, as others have also found, that self-related items may capture attention more, and that this may be a mechanism contributing to the enhanced memory for self-related items⁵.”

p. 12, line 291:

“While mirror self-recognizers attended longer to the self-assigned than the other-assigned objects during the object encoding phase, mirror non-recognizers did not show preferential attention to either of the objects at encoding. Yet, mirror non-recognizers later showed better memory of other-assigned than self-assigned objects. Thus, while attention to the object may provide a mechanism for better memory for self-assigned objects in the recognizers, it doesn’t capture non-recognizers’ enhanced memory for other-assigned objects. In previous work documenting a bias to remember the targets of others’ attention in 8-month-old infants, we

found that memory was greatest in infants who spent more time attending to the agent rather than the object³², perhaps because altercentric encoding relies on attention between agent and object, whereas self-related encoding does not.“

3. Related to the above, and also an independent concern: to what degree was coding done of the experimental procedure to ensure that self vs. other items were introduced in the same manner? It is important to undertake some coding to this end because obviously the experimenter could not be blind to which toys were the infants' versus puppets' toys.

We have now coded the duration that the objects were presented to the infants during encoding time, and there was no difference in presentation time between the self-assigned and other-assigned objects (A Bayesian Wilcoxon signed rank test yielded moderate evidence against a difference: $BF = 0.138$, $W = 1337$, $Rhat = 1$). We thank the reviewer for this suggestion and have now added this information to the Methods section (**p. 14, Line 355**).

In addition, we would like to point out that the experimenter was blind to the infants' mirror status (see details under the next comment); thus, she could not have adapted her presentation of self- and other-assigned objects depending on mirror status. As we find opposing patterns in mirror recognizers and non-recognizers, infants' differential looking in the test phase can therefore not be explained by systematic differences in the way self-assigned versus other-assigned objects would have been presented (as this would have had to differ systematically between mirror recognizers and non-recognizers as well).

This is additionally supported by experiment 2 where the presentation of the self- versus other-assigned objects was automated and therefore identical. Nevertheless, the same looking pattern observed in the mirror recognizers was replicated, indicating that this pattern could not have resulted from differences in presentation style.

Similarly, was the experimenter aware of infants' self-recognition status?

We thank the reviewer for pointing out that we had neglected to report this information. From the $N = 73$ infants in experiment 1, $N = 29$ infants did the mirror test after the self-recognition test, so the experimenter could not have known about their mirror status. The other $N = 44$ infants did the mirror test first, but on a different day as part of a different study (where the mirror task was also acquired). The experimenter did not check the infants' mirror status before conducting the self-reference task and can therefore be assumed to have been blind towards their mirror status. Note that there was no evidence for an effect of task order on the reported results. We have added this information to the Methods (**p. 15, line 402**).

4. In the discussion, I'd love to see more about the potential utility of the altercentric bias. I find the current writing to be a bit hand-wavy.

We have now added some more discussion about the potential utility of the altercentric bias:

p. 11, line 265:

Although the utility of such an altercentric bias is necessarily speculative, it may facilitate infant learning by biasing attention to, and encoding of, the targets of others' attention and action. This would allow infants to prioritize learning about contents, which have already been curated by more knowledgeable others.

5. Throughout there are places in which the wording implies that infants' memory for self-associated objects versus other associated objects were directly compared against one another (i.e., on test, shown self versus other objects and their preference recorded). I'd suggest clarifying throughout the manuscript.

We thank the reviewer for pointing out this unclarity. We have changed the wording where it might be misleading in the manuscript.

Relatedly, this wording made me wonder why the authors didn't take this approach.

In our study, we were interested in testing whether infants remembered the objects. Had we directly compared self- and other-assigned objects, we may have revealed a preference for one over the other, but this would not have allowed us to conclude anything about their memory of the objects. It is with the novelty/familiarity preference for the previously seen compared to a novel object that we can test whether they remember the previously seen object.

In summary, I think this manuscript is an interesting attempt to directly test the altercentric stance perspective, and tie the erosion of this stance to the emergence of a sense of self. However, more work needs to be done to convince the reader that the findings reflect what the authors believe they reflect. While my comments may seem critical, I'd like to underscore that I raise these concerns, and hope the authors will address them, because I think that if the initial altercentric stance theory is correct it will revolutionize the field! For these reasons, I want to ensure that this novel and important theory is tested in the most rigorous of ways.

We would like to thank the reviewer for their critical and constructive comments that have strongly helped us to improve our manuscript.